# Co-Delivery of Cisplatin and Gemcitabine via Viscous Nanoemulsion for Potential Synergistic Intravesical Chemotherapy

**DOI:** 10.3390/pharmaceutics12100949

**Published:** 2020-10-07

**Authors:** Ting-Yu Chen, Ming-Jun Tsai, Li-Ching Chang, Pao-Chu Wu

**Affiliations:** 1School of Pharmacy, Kaohsiung Medical University, Kaohsiung City 807, Taiwan; sylviab074@gmail.com; 2Department of Neurology, China Medical University Hospital, Taichung City 404, Taiwan; d22570@mail.cmuh.org.tw; 3School of Medicine, Medical College, China Medical University, Taichung City 404, Taiwan; 4School of Medicine for International Students, I-Shou University, Jiaosu Village Yanchao District, Kaohsiung City 82445, Taiwan; 5Department of Medical Research, Kaohsiung Medical University Hospital, Kaohsiung City 807, Taiwan; 6Drug Development and Value Creation Research Center, Kaohsiung Medical University Hospital, Kaohsiung City 807, Taiwan

**Keywords:** cisplatin, gemcitabine, dual-loaded microemulsion, intravesical instillation

## Abstract

Combined chemotherapy is an effective and safe treatment for cancers. Co-administration of cisplatin and gemcitabine produces a synergistic effect for bladder cancer treatment, so viscous microemulsions were developed for co-delivery of cisplatin and gemcitabine to extend the retention time and improve the permeability of chemotherapeutic drugs into the urothelium by intravesical administration. Results showed that the deposition amounts of cisplatin and gemcitabine significantly increased in in vitro and in vivo study. The penetration depth in bladder tissue samples increased from 60 to 120 μm. The dual-loaded formulation also showed thermodynamic and chemical stability, demonstrating that these gel-based microemulsions are promising drug delivery carriers for chemotherapy agents by intravesical administration.

## 1. Introduction

Cisplatin (Pt(NH_3_)_2_Cl_2_, MW 301.1, log P −2.19, pka 5.1, and solubility 2.53 mg/mL) is a well-known chemotherapeutic agent used for the treatment of numerous cancers including bladder, lung, ovarian, and testicular cancers. In cells, it crosslinks with the purine bases on DNA, thereby interfering with DNA repair mechanisms, resulting in DNA damage, and then induces apoptosis [1]. Unfortunately, platinum responsiveness is high primarily, but many cancer patients will eventually relapse with cisplatin-resistant disease and numerous undesirable side effects. Resistance mechanisms have been proposed including changes in cellular uptake and efflux, biotransformation and detoxification in the liver, inhibition of apoptosis, and increased DNA repair [2]. Hence, combination therapies of cisplatin with other drugs have been highly considered to overcome drug resistance and reduce toxicity [2,3,4]. Gemcitabine hydrochloride (C_9_H_11_F_2_N_3_O_4_, MW 263.2, logP −1.4, pka 3.6, and water solubility of 51.3 mg/mL) is a chemotherapy medication used to treat a number of types of cancer including bladder cancer, ovarian cancer, non-small cell lung cancer, pancreatic cancer, and breast cancer [5]. After entering the cell, gemcitabine becomes attached to phosphate to become thrice-phosphorylated; then, it is incorporated into new DNA strands, creating an irreparable error that leads to the inhibition of further DNA synthesis, thereby leading to cell death [5,6,7]. Numerous studies have shown that the combination of cisplatin and gemcitabine yields a synergistic anti-tumor effect—in particular, with a ratio of 1/10~1/16, exhibiting better performance [3,8,9,10,11].

Intravesical therapy is an important strategy after transurethral tumor resection of visible tumors for superficial bladder cancer treatment, because it can cause high local drug concentration and low systemic drug concentration [12,13,14,15]; however, there are challenges in intravesical treatment, such as the barrier effect of the urothelium and the rapid decrease in drug concentration with urination [7,16]. Extending the retention time in the bladder and improving the permeability of chemotherapeutic drugs into the urothelium are important tasks for effective intravesical administration. In the past few decades, nanoparticles have been developed to improve the permeability [17,18,19,20,21]. Hydrogels have been used to prolong the retention time of the load in the bladder and sustainably release drug(s) to enhance local drug concentration at disease sites [22,23,24,25]. Microemulsions are thermodynamically stable, isotropically active colloids that are composed of oil, water, surfactants, and cosurfactants. Their viscosities are low and droplet sizes are in the submicron range. As drug delivery carriers, they offer high drug solubilizing capacity, ease of preparation, long-term stability, and capability to increase permeability for hydrophobic and hydrophilic drugs [17,19,26,27]. In this study, a viscous microemulsion was used as a drug delivery carrier to simultaneously increase the penetration efficiency of gemcitabine and cisplatin into the bladder endothelial tissue.

## 2. Materials and Methods

### 2.1. Materials

Gemcitabine hydrochloride was purchased from Scinopharm (Tainan, Taiwan), cisplatin, acetaminophen, paraformaldehyde carbamic acid ethyl ester (urethane), nickel chloride, methylcellulose, chloral hydrate, and rhodamine B from Sigma-Aldrich (St. Louis, MO, USA), while diethylene glycol monoethyl ether (Transcutol) was obtained from Fluka (Forest parkway, GA, USA). Benzalkonium chloride and perchloric acid were obtained from Merck Chemicals (Darmstadt, Germany); tetrahydrouridine was purchased from Calbiochem (San Diego, CA, USA); capryol 90 (propylene glycol monocaprylate), sodium diethyldithio carbamate, cetrimonium bromide, and 1,5-pentanediol were obtained from Alfa Aesar (Ward Hill, MA, USA), and pentane sulfonic acid was obtained from Wako (Osaka, Japan). All other chemicals and solvents were of analytical reagent grade.

### 2.2. Animals

Female Sprague–Dawley (SD) rats weighing 150–200 g purchased from BioLASCO Taiwan Co., Ltd. (Taipei, Taiwan) were used for this study. The animals were housed at an ambient standard temperature, with a 12-h day/night cycle, and fed a standard pellet diet and water. The protocol of all experiments was carried out in accordance with the Code of Ethics of the World Medical Association and approved by the Experimental Animal Welfare and Ethics Committee of Kaohsiung Medical University (no. 104144).

### 2.3. Drug-Loaded Formulation Preparation

Drug-loaded microemulsion formulations were prepared by mixing oil phase with the mixture of surfactant and cosurfactant, and then water was added precisely into oily phases with vortex for 2 min at 25 °C to obtain a homogenous mixture. Then, methylcellulose as thickener was incorporated into the mixture and magnetically stirred overnight to obtain viscous formulations. The compositions of formulations without model drugs are listed in Table 1. Finally, model drugs of 1% gemcitabine, 0.1% cisplatin, and/or 0.05% rhodamine B were dissolved in the viscous mixture.

### 2.4. Characterization of Microemulsion

The droplet size distribution and polydispersity index (PDI) of drug-loaded formulations were determined by photon correlation spectroscopy (Zetasizer ZS, Malvern Instruments, Ltd., Malvern, UK). Measures were always performed at ambient temperature in triplicate.

Viscosities of drug-loaded formulations were measured by Brookfield type viscometer (Model LVDV-II, USA). Samples were loaded into the cone-plate and then heated and maintained at 37 °C for 1 min by a thermostatic pump. The viscosities of formulations were determined at rotation rate of 120.0 rpm and recorded after 30 s.

The morphology of the drug-loaded microemulsion formulation was determined using transmission electron microscopy (Hitachi 7700, Tokyo, Japan). Briefly, one drop of sample was deposited on a film-coated copper grid, followed by staining with 2% phosphotungstic acid aqueous solution, and then the sample was allowed to dry at ambient temperature before examination.

### 2.5. In Vitro Permeation Studies

In vitro permeation study paradigm delivery of gemcitabine and cisplatin was investigated by using vertical static Franz diffusion cells. The effective diffusion area provided in this set-up was 3.46 cm^2^. Cells were washed and filled with 20 mL of pH 7.4 citric acid–sodium phosphate solution to maintain sink conditions. The receptor compartment was maintained at 37 °C using a recirculating water bath system. At predetermined times of 0.5, 1, 2, 3, 4, and 6 h, receptor fluid of 1 mL was withdrawn, and then the same volume of fresh fluid was replaced into the receiver compartment. The transdermal amounts of gemcitabine and cisplatin were quantified using modified high performance liquid chromatography (HPLC) [20,28]. Each experimental formulation was measured in triplicate and the mean value was presented.

After 6 h, the residual drug on the surface of applied tissue was washed with water for three times. The drug-accumulated amount in the biological membrane was extracted with 4 mL of the receptor buffer by horizontal shaking overnight, the resulting solution was centrifuged at 4000 rpm for 30 min, and the gemcitabine and cisplatin levels were then analyzed.

### 2.6. In Vivo Intravesical Administration of Dual-Loaded Formulations

Female SD rats were catheterized transurethrally under anesthesia with a PE 50 tube. Urine was removed by slightly pressing the lower abdomen, and then one milliliter of dual-loaded microemulsion formulations with and without polymer and control aqueous solution containing 1% gemcitabine, 0.1% cisplatin, and 0.05% rhodamine B was then pipetted into the hub of the catheter and instilled into the rat bladder cavity. The urethral orifice was tied with cotton thread to prevent drug-loaded formulation leakage. After 1 h incubation, the animal was sacrificed immediately (scheme 1) or 1 h later (scheme 2), with bladder and blood samples being taken for subsequent tests.

### 2.7. In Vivo Confocal Laser Scanning Microscopy (CLSM) Analysis

To clarify the formulations located in different urothelial layers of the bladder, rhodamine B was used as fluorescent marker in both viscous microemulsion and control solution. On ending the intravesical administration, bladder tissues were examined for rhodamine B fluorescence images by CLSM through the z-axis at ca. 20 μm increments (FV 500, Olympus, Tokyo, Japan). Optical excitation was carried out with a 500 nm argon laser beam, and the fluorescence emission was detected at 540 nm.

### 2.8. Irritation Test

The treated and untreated bladder area samples were collected after intravesical administration and stored in 10% formalin solution in phosphate buffer solution at least for 24 h. These samples were then dehydrated using ethanol, embedded in paraffin, stained with hematoxylin and eosin, then finally observed under Nikon light microscope (Nikon Eclipse Ci, Tokyo, Japan) and compared with the control group for any irritation.

### 2.9. Data Analysis

Data were presented as mean ± standard deviation. Statistical analysis was performed using a computer program, SPSS-Statistic software 22.0 (IBM Corp, Armonk, NY, USA). Differences among groups were tested using analysis of variance with the Turkey’s post-hoc test and were considered statistically significant if the *p*-value was <0.05.

## 3. Results and Discussion

### 3.1. Physicochemical Characteristics

The mean droplet sizes of the prepared gemcitabine-loaded microemulsions without methylcellulose ranged from 9.47 to 26.73 nm, and the polydispersity index (PDI) ranged from 0.06 to 0.30, indicating that nanoscale droplet sizes with relative monodispersing colloids were obtained from the following formulations 1 to 4 (Table 2). After thickener agent methylcellulose was incorporated, the gel-based microemulsion droplets of formulae F3 obtained were spherical and relatively even in size (Figure 1). The droplet size increased to 200~250 nm. In 2007, Su and coauthors reported that nanoemulsions with median size of around 200 nm exhibited moderate transdermal delivery effects [29]. Tuan-Mahmood and coauthors [30] also reported that only particles with a size of 50 to 500 nm were able to penetrate physiological membranes. This implied that the microemulsion formulations had potential for drug delivery through biological membranes. The viscosity of gel-based microemulsions ranged from 410.27 to 1104.00 cps. A trend of viscosity increase was observed when the content of benzalkonium chloride increased or 1,5-pentanediol decreased in the formulations.

### 3.2. In Vitro Permeation and Drug Deposition Evaluation

The drug cumulative amounts of different gemcitabine-loaded microemulsion formulations were plotted against time, as shown in Figure 2. It was found that viscous microemulsion formulations significantly increased the permeability of gemcitabine when compared to the control solution. In comparison, for the effect of formula composition on drug penetration, it could be seen that the drug permeability from the formulation containing 0.5% benzalkonium chloride and 5% Tween 80 was lower than that of the formulation containing 1.5% benzalkonium chloride (F1 vs F2), indicating that benzalkonium chloride is a strong emulsifier and penetration enhancer that could help microemulsion formation and remarkably enhance drug permeability at a very low concentrations [27,31].

The permeation enhancement decreased as the amount of benzalkonium chloride increased from 1.5 to 2.0% (F2 vs. F3). The result was consistent with previous findings of the amount of surfactant in the microemulsion formulation increasing, the thermodynamic activity of drug in the formulation decreases, leading to a shrinkage in the drug permeation rate [32,33]. In addition, decreasing the amount of 1,5-pentanediol from 20 to 15% significantly increased the flux. The phenomenon may be due to the excess amount of cosurfactant incorporated; reducing the drug thermodynamic activity in the microemulsion resulted in reduced permeation rate [34].

Previous studies reported that combinations of cisplatin and gemcitabine could produce synergistic anti-tumor effects; hence, the experimental formulations were dual-loaded with 1% gemcitabine and 0.1% cisplatin and studies were conducted [10,11]. As shown in Table 3, the flux and deposition amount of gemcitabine and cisplatin were significantly increased by using microemulsions as carriers, especially the hydrophilic compound of gemcitabine.

### 3.3. In Vivo Intravesical Application

The drug plasma concentrations and depositions in bladder tissue after intravesical administration of dual-loaded microemulsion and control solution are presented in Figure 3. For the plasma concentration, there was no significant difference among the control, formulation with polymer, and formulation without polymer at scheme 1 and scheme 2, indicating that neither of the experimental formulations caused more serious systemic side effects. It was also found that cisplatin was eliminated faster than gemcitabine in the plasma.

For drug deposition in bladder tissue, it was found that dual-microemulsions with or without polymer could increase deposition amounts of gemcitabine and cisplatin, showing that microemulsions could significantly enhance the permeability of hydrophilic and hydrophobic drugs. The results are consistent with the above in vitro penetration results, indicating that nanoparticles can promote the permeability of chemotherapeutic drugs into the urothelium [17,19,20]. In scheme 1 (0 h after 1 h incubation), the deposition amount of gemcitabine and cisplatin of the formulation without polymer was higher than that of the formulation with polymer, indicating that the diffusion rates of gemcitabine and cisplatin decreased via increasing the viscosity of formulations. On the contrary, in scheme 2 (1 h after 1 h incubation), the deposition amount of both drugs of the formulation without polymer was higher than that of the formulation without polymer, indicating that the high viscosity emulsion could control the release while also reducing the excretion of drugs. The result is in agreement with previous studies that reported that adhesive or viscous formulations such as hydrogel could prolong the retention time of the load in the bladder and sustain drug release, resulting in improved therapeutic efficacy [16,24,35].

### 3.4. In Vivo CLSM Analysis

To elucidate the bladder penetration behavior of dual-loaded microemulsions, the CLSM method was conducted. Confocal images were acquired after both microemulsion and control groups had been applied to the bladder cavity of SD rats for 1 h incubation. Figure 4 shows the results for microemulsion in the CLSM studies. It was found that the control solution group (drugs dissolved in water) presented a weak rhodamine B signal and the delivery depth was only around 60 μm. By contrast, the microemulsion showed a significantly higher intensity of rhodamine B than the control group and the penetration depth was up to 120 μm.

### 3.5. Irritation Test

At the end of the intravesical instillation of different formulations including drug solution, drug-loaded microemulsion, and unloaded microemulsion, the bladder tissues were sectioned for histopathological examination to further verify the irritation caused by the formulations. As shown in Figure 5A, sections of untreated bladder (control) showed well-defined epidermal and dermal layers. The basement membrane was also well-formed and presented just adjacent to the topmost layer of the epidermis. In the case of the drug solution, some epithelial cells with exfoliation were observed, indicating that the chemotherapy agent had irritant properties (Figure 5B). In the case of drug-loaded and unloaded microemulsions (Figure 5C,D), no significant irritation was found, indicating that the experimental microemulsion presented little change and even diminished the skin irritation caused by drugs. The result is consistent with previous studies [36,37] that reported that microemulsions could not cause irritation, although microemulsions contain large amounts of surfactants and cosurfactants.

### 3.6. Stability

No phase separation was observed when samples were subjected to centrifugation tests of 3500 rpm for 30 min and 10,000 rpm for 5 min, showing that the formulation possessed thermodynamic stability. Storage of the dual-loaded formulation for 2 months at 30 °C showed no change in the appearance of the system; the droplet size changed from 924 ± 23 to 899 ± 13 nm, the viscosity changed from 946 ± 17 to 90515 cps, and the drug content of the formulation remained within 93.43–99.25%. This result demonstrates that the dual-loaded formulation exhibited both chemical and physical stability throughout the storage period.

## 4. Conclusions

An intravesical delivery system of gel-based microemulsion was established for the co-delivery of gemcitabine and cisplatin, which achieved superior drug permeability to that of the compared aqueous solution in in vitro permeation and in vivo intravesical administration. In irritation tests, no significant irritation was observed after intravesical administration. The dual-loaded formulation exhibited both chemical and physical stability throughout a two-month storage period, demonstrating that the gel-based microemulsion was a promising drug delivery carrier for chemotherapy agents by intravesical administration.

## Figures and Tables

**Figure 1 pharmaceutics-12-00949-f001:**
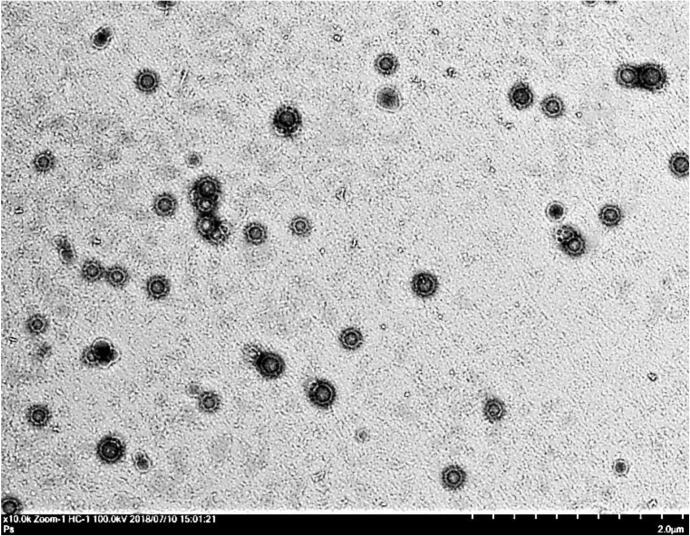
Transmission electron microscopy image of gemcitabine-loaded microemulsion of F3.

**Figure 2 pharmaceutics-12-00949-f002:**
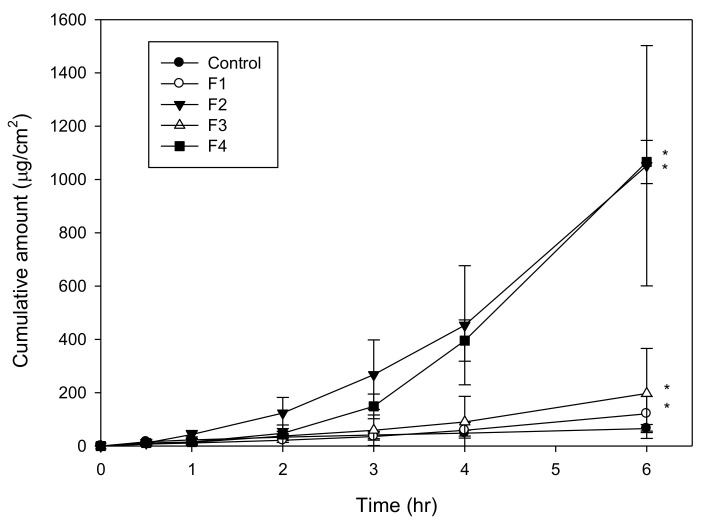
In vitro penetration time profiles of gemcitabine-loaded formulations and control drug solution (* *p* < 0.05).

**Figure 3 pharmaceutics-12-00949-f003:**
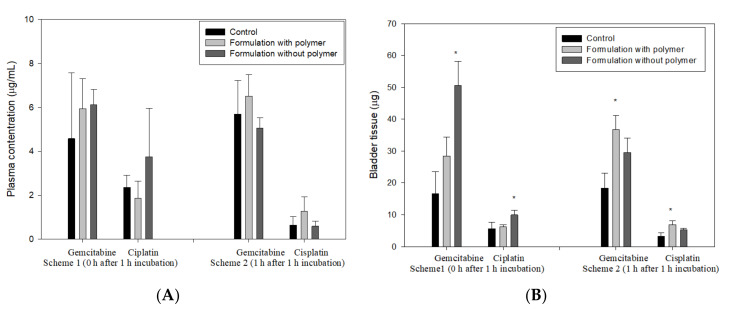
The plasma concentration (**A**) and deposition amount (**B**) of gemcitabine and cisplatin in the bladder of dual-loaded microemulsions of F3 and control solution after intravesical administration (scheme 1: 0 h after 1 h incubation; scheme 2: 1 h after 1 h incubation) (* *p* < 0.05).

**Figure 4 pharmaceutics-12-00949-f004:**
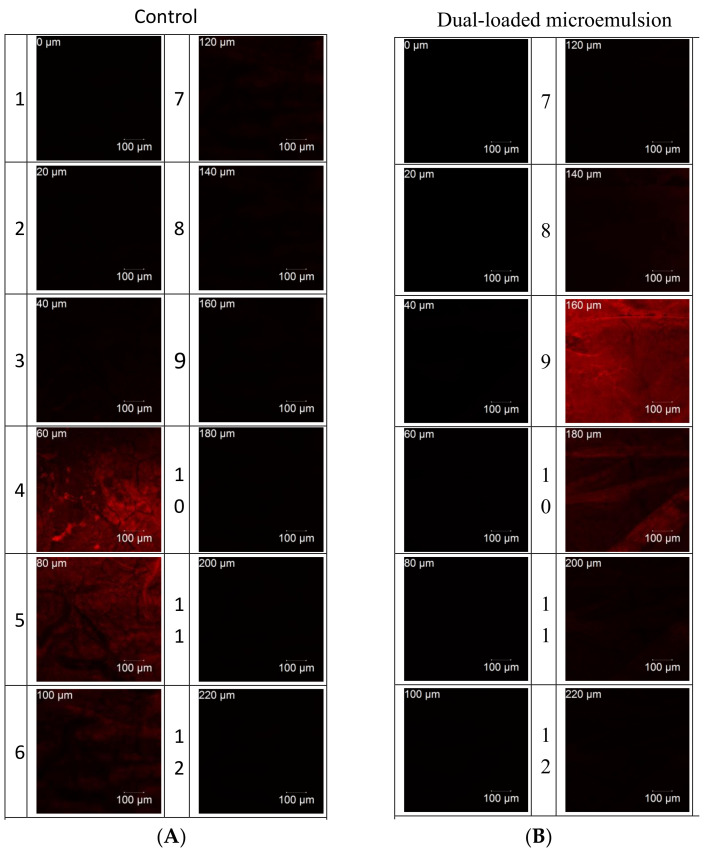
Confocal Laser Scanning Microscope micrographs of bladder tissue after in vivo application of dual microemulsion for 1 h. (**A**) Aqueous control solution; (**B**) dual-loaded microemulsion of F3 containing 1% gemcitabine, 0.1% cisplatin, and 0.1% rhodamine. Numbers (1~12) indicate the number of layers, each layer is 20 μm.

**Figure 5 pharmaceutics-12-00949-f005:**
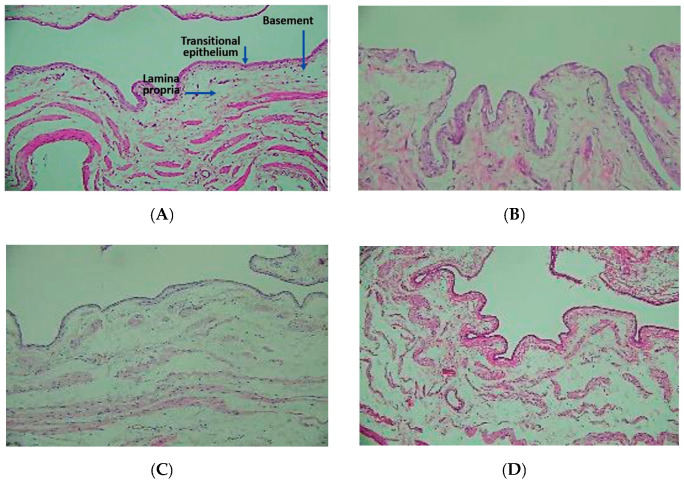
Histopathological images of bladder after treatment with different formulations (100×). (**A**) Untreated, (**B**) Drug solution treated, (**C**) Dual-loaded microemulsion treated and (**D**) Un-loaded microemulsion treated.

**Table 1 pharmaceutics-12-00949-t001:** Composition of formulations.

Formulation (g)	F1	F2	F3	F4
Capyrol	5.0	5.0	5.0	5.0
Benzalkonium chloride	0.5	1.5	2.0	2.0
Tween 80	5.0			
1,5-Pentanediol	20.0	20.0	20.0	15.0
Carbitol	10.0	10.0	10.0	10.0
Water	59.5	63.5	63.0	68.0
Methylcellulose	2.0	2.0	2.0	2.0

**Table 2 pharmaceutics-12-00949-t002:** Droplet size, polydispersity index (PDI), and viscosity of drug-loaded microemulsions.

Formulation	Droplet Size nm	PDI	Viscosity Cps
F1	10.26 ± 0.15	0.06 ± 0.00	410.27 ± 1.16
F2	12.50 ± 0.10	0.14 ± 0.01	687.90 ± 3.16
F3	14.13 ± 0.42	0.16 ± 0.09	923.93 ± 1.61
F4	26.73 ± 0.06	0.30 ± 0.01	1104.00 ± 4.70

**Table 3 pharmaceutics-12-00949-t003:** Flux, deposition, and enhancement ratio (ER) of dual-loaded formulations and control group (C).

	Gemcitabine				Cisplatin			
Formulation	Flux_G_ (μg/cm^2^/h)	ER_GF_	Deposition (μg/cm^2^)	ER_GD_	Flux (μg/cm^2^/h)	ER_CF_	Deposition (μg/cm^2^)	ER_CD_
C	4.84 ± 0.64	1.0	0.57 ± 0.24	1.0	1.30 ± 0.04	1.0	0.05 ± 0.09	1.0
F3	92.52 ± 62.93 *	19.0	30.95 ± 7.08 *	54.0	24.67 ± 3.37 *	18.0	0.91 ± 0.30 *	18.0
F4	113.69 ± 27.38 *	23.5	40.36 ± 4.48 *	70.0	17.49 ± 4.09 *	13.0	1.35 ± 0.94 *	27.0

ER_GF_: Gemcitabine Flux_(control)_/Gemcitabine Flux_(formulation)_, ER_GD_: Gemcitabine Deposition_(control)_/Gemcitabine Deposition_(formulation),_ ER_CF_: Cisplatin Flux_(control)_/Cisplatin Flux _(formulation)_, ER_CD_: Cisplatin Deposition_(control)_/Cisplatin Deposition_(Formulation)_, *: significant difference.

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
