# Peer review of "Co-Delivery of Cisplatin and Gemcitabine via Viscous Nanoemulsion for Potential Synergistic Intravesical Chemotherapy"

_pharmaceutics, 2020, doi:10.3390/pharmaceutics12100949_

Round 1

Reviewer 1 Report

In this work cisplatin and gemcitabine anticancer drugs in nanoemulsions are co-delivered in intravesical chemotherapy. The work has got interesting findings, however, a major revision is adviced.

Major remarks:

  1. The nanoemulsion preparation is not clearly described, e.g. in page 2, line 89: What does it mean: 'incorporated into the mixture overnight slowly? The method should be reproducible, thus it needs a clear description. line 91: ‘…0.1% cisplatin and/or 0.05% rhodamine B were dissolved in the viscous mixture…’ In which formulation was rhodamin B with cisplatin or instead of cisplatin added? What is the unit of components in Table 1?
  1. Figure 2: There is not a significant difference between F2 and F3 composition. What is the explanation of huge difference in the in vitro penetration data? The present explanation is not convincing.
  2. Section 3.3. It is not clear what do authors mean under 'formulation with polymer and without polymer'. It must be described in the experimental section.
  3. Page 6, line 199: ‘For drug deposition in bladder tissue, it was found that dual-microemulsions with or without polymer could both increase deposition amounts of gemcitabine and cisplatin…’ In contrast, cisplatin increase was smaller in bladder tissue than in plasma.
  4. Section 3.6: For stability test, size distribution of microemulsion after 2 months storage must be provided.

Minor comments:

  1. Page 1, line 23: ‘produce’ must be corrected to ‘produces’
  2. Page 1, line 29: ‘…thermodynamic stability and chemical satiability…’ should be corrected to ‘…thermodynamic- and chemical stability …’
  3. Page 2, line 53: ‘…treatment because of it…’ should be changed to ‘…treatment, because it…’
  4. Section 2.5: Is it ex vivo or in vitro, the title indicates it differently from the first sentence.
  5. Page 3, line 123: ‘then’ should be deleted, since it was repeated in the sentence.
  6. Page 4, line 150: Table 2 should be written instead of Table 1.
  7. Page 4, line 151: Fig.1 should be written instead of Fig.2.
  8. Page 5, line 182: Table 3 should be written instead of Table 2.

Author Response

Thank you for your comments

The original manuscript has been revised in accordance with your suggestions 

Reviewer 2 Report

Authors show a vehicle for increasing the penetrability of a combined therapy of cisplatin and gemcitabine in the bladder. In general terms both, dual therapy and the use of gels or polymers have already been described in bladder cancer, with recent publications not cited in the introduction (29997433; Nano Research volume 12, pages1389-1399(2019); DOI: 10.1158/1078-0432.CCR-17-1082). In this sense the article does not bring enough novelty for a Q1 journal in the field.

Throughout the article there are some important issues and some minor ones that are worth mentioning:

Important issues:

-The in vivo experiments are performed at very short times to know how long the drug really remains in the epithelium. In addition, the in vitro penetration experiments (Fig. 2) indicate that in the first two hours there is hardly any penetration, this is more evident at 6 hours.

-There are no significant differences nor specified in Fig.2

-It is not specified what the control consists of until section 3.4. It must be specified the first time it is mentioned and also in the material and methods.

-Section 3.3 does not specify which polymer is removed and why.

-In figure 3 it should be shown that the differences in bladder in the two schemes are not due to the fact that the formulation without polymer remains on the surface without penetrating into scheme 1 and therefore in scheme 2 most of therapy has already been lost. It should be explained why the more viscous emulsions facilitate penetration, when one would expect the opposite.

-In the stability study it would be necessary to measure again viscosity and efficiency of the vehicle.

- To really show an improvement over  combined therapy without microemulsions an efficacy in vivo experiment should be performed on a bladder cancer model mouse model.

Minor issues:

Figure 1 does not say in which formulation the microemulsion is, nor is it clarified in figures 3, 4 and 5 with which formulation the experiments have been done.

In line 151 where it says Fig. 2 it should say Table 2

Line 176, change additional by addition.

Line 182, change Table 2 by Table 3

Line 219, change rohadamine by rhodamine

In figure 5, indicate with arrows the keratin layer, the epidermis and the dermis

Author Response

Thank you for your comments

The original manuscript has been revised in accordance with your suggestions.

Round 2

Reviewer 1 Report

The corrections have been made according to my comments. Two tiny errors should be still corrected.

In the abstract there is a mistyping: 'satiabilty' instead of 'stability'.

In section 2.4 the name of instrument for photon correlation spectroscopy is missing (probably Zetasizer ZS).

Reviewer 2 Report

With the corrections of version 2 the article could be accepted